# Pooled prevalence and associated factors of chronic undernutrition among under-five children in East Africa: A multilevel analysis

Getayeneh Antehunegn Tesema[1]*, Yigizie Yeshaw[1,2], Misganaw Gebrie Worku[3], Zemenu Tadesse Tessema[1], Achamyeleh Birhanu Teshale[1]

1 Department of Epidemiology and Biostatistics, Institute of Public Health, College of Medicine and Health Science, University of Gondar, Gondar, Ethiopia, 2 Department of Human Physiology, School of Medicine, College of Medicine and Health Science, University of Gondar, Gondar, Ethiopia, 3 Department of Human Anatomy, School of Medicine, College of Medicine and Health Science, University of Gondar, Gondar, Ethiopia

* getayenehantehunegn@gmail.com

## Abstract

### Background

Childhood undernutrition is the leading cause of under-five mortality and morbidity in the world particularly in East African countries. Although there are studies on child undernutrition in different East African countries, our search of the literature revealed that there is limited evidence of a pooled analysis of these studies. Therefore, this study aimed to investigate the pooled prevalence and associated factors of chronic undernutrition (i.e. stunting) among under-five children in East Africa.

### Methods

A pooled analysis of the Demographic and Health Surveys (DHSs) in 12 East African countries was conducted. A total weighted sample of 79744 under-five children was included in the study. Mixed-effect logistic regression analysis was used to identify significant factors associated with chronic undernutrition since the DHS data has a hierarchical structure. The intra-class correlation coefficient (ICC), Median Odds Ratio (MOR), Likelihood Ratio (LR)-test, and deviance was used for model comparison. Variables with p-value <0.2 in the bivariable mixed-effect logistic regression analysis were considered for the multivariable analysis. In the multivariable multilevel analysis model, the Adjusted Odds Ratio (AOR) with the 95% Confidence Interval (CI) were reported for significant factors.

### Results

The pooled prevalence of chronic undernutrition among underfive children in East Africa was 33.3% (95% CI: 32.9%, 35.6%) ranging from 21.9% in Kenya to 53% in Burundi. Children whose mothers lived in rural area (AOR = 1.11, 95% CI: 1.06, 1.16), born to mother who had no formal education (AOR = 1.42, 95% CI: 1.34, 1.50) and primary education (AOR = 1.37, 95% CI: 1.31, 1.44), being in poor household (AOR = 1.66, 95% CI: 1.58, 1.74), and

**Data Availability Statement:** The data used in this study are from the Measure DHS program (www.dhsprogram.com) and can be accessed following the protocol outlined in the Methods section.

**Funding:** The authors received no specific funding for this work.

**Competing interests:** The authors have declared that no competing interests exist.

**Abbreviations:** ANC, Antenatal Care; AOR, Adjusted Odds Ratio; CI, Confidence Interval; DHS, Demographic Health Survey; GLMM, Generalized Linear Mixed Models; ICC, Intra-class Correlation Coefficient; LLR, log-likelihood Ratio; LR, Likelihood Ratio; MOR, Median Odds Ratio; SSA, Sub-Saharan Africa; WHO, World Health Organization.

middle household (AOR = 1.42, 95% CI: 1.35, 1.49), child aged 36–48 months (AOR = 1.09, 95% CI: 1.04, 1.14), being male (AOR = 1.19, 95% CI: 1.15, 1.23), 2nd - 4th birth order (AOR = 1.08, 95% CI: 1.03, 1.13), and above 4th 1.27 (AOR = 1.27, 95% CI: 1.19, 1.35), home delivery 1.09 (AOR = 1.09, 95% CI: 1.05, 1.13), small size at birth (AOR = 1.35, 95% CI: 1.29, 1.40) and being multiple births (AOR = 1.98, 95% CI: 1.81, 2.17) were associated with increased odds of stunting. While, antenatal care visit (AOR = 0.89, 95% CI: 0.86, 0.93), mothers aged 25–34 (AOR = 0.83, 95% CI: 0.79, 0.86) and ≥ 35 years (AOR = 0.76, 95% CI: 0.72, 0.81), large size at birth (AOR = 0.85, 95% CI: 0.81, 0.88), and family size >8 (AOR = 0.92, 95% CI: 0.87, 0.98) were associated with decreased odds of stunting.

## Conclusion

The study revealed that stunting among under-five children remains a major public health problem in East Africa. Therefore, to improve child nutrition status the governmental and non-governmental organizations should design public health interventions targeting rural residents, and the poorest households. Furthermore, enhancing health facility delivery, ANC visit, and maternal health education is vital for reducing child chronic undernutrition.

## Background

Adequate nutrition is vital for healthy growth, and neurological and cognitive development in early childhood [1, 2]. The first 1000 days after conception are the critical period for infant growth and development [3]. Stunting is a linear growth failure that causes both physical and cognitive delays in growth and development [4] and it is the best predictor of chronic child undernutrition and household well-being [5]. Stunting is defined as the inability to attain potential height for age and under-five children are particularly vulnerable to malnutrition [6]. A child is considered to be stunted when their height for age is minus two standard deviations below the median population [7].

Malnutrition is the leading cause of morbidity and mortality in under-five children [8]. Globally, an estimated 178 million under-five children are stunted, of whom more than 90% live in Africa and Asia [9]. It affects one-third of under-five children in low and middle-income countries [10]. The prevalence of stunting among under-5 children is highest in Africa (36%), of these the largest number of stunted children is found in East Africa [11]. More than half (54%) of under-five mortality each year is due to malnutrition and its related complication [12].

It is well known that childhood malnutrition has significant consequences on child growth and development [13, 14]. It has both short and long-term adverse effects [15]. The short term consequences are recurrent infections [16], increased diseases severity [17], and delayed recovery from the disease [18], delayed physical and mental development [19], whereas the long term consequences of stunting are poor academic performance [20], premature death [14] and increased the risk of chronic diseases such as Diabetic Mellitus (DM), Hypertension (HTN) and heart disease [21, 22]. In addition, malnutrition has a significant negative consequence on the future reproductive health of female children [23, 24].

Previous studies on stunting among under-five children revealed that the causes of stunting are multifactorial [25–28]. Commonly reported predictors of stunting among under-five children include residence, duration of breastfeeding, wealth status, media exposure, maternal education, husband education, Antenatal Care (ANC) visit during pregnancy, Postnatal Care

(PNC), place of delivery, health care access, women decision making autonomy, childhood illness, congenital diseases, maternal Body Mass Index (BMI), birth order and family size [29–33].

According to UNICEF data, the global progress of stunting has declined from 255 million to 159 million from 1990 to 2014 [34]. Nevertheless, today, one in four under-five children in East African countries is stunted [35]. Despite the Sustainable Development Goal (SDG) ambitious target to reduce stunting among children by 40% by 2025 [36, 37], the prevalence in East Africa remains unacceptably high [38, 39].

Though there are studies conducted on stunting in East African countries [30–32, 40, 41], these were mainly based on hospital records, and are unable to capture the pooled prevalence and associated factors of stunting among under-five children in East Africa. Therefore, this study aimed to investigate the pooled prevalence and associated factors of chronic undernutrition among under-five children in East Africa. The study findings could help to inform the design of evidence-based public health interventions for reducing childhood undernutrition, and subsequently reduce child mortality, in East Africa.

## Methods

### Data source

This study was based on the most recent Demographic and Health Surveys (DHS) conducted in the 12 East African countries (Burundi, Ethiopia, Comoros, Uganda, Rwanda, Tanzania, Mozambique, Madagascar, Zimbabwe, Kenya, Zambia, and Malawi). These datasets were merged to determine the pooled prevalence and associated factors of stunting among under-five children in East Africa. The DHS is a nationally representative survey that collects data on basic health indicators like mortality, morbidity, family planning service utilization, fertility, maternal and child health. The data were derived from the *https://dhsprogram.com/data/available-datasets.cfm*. The DHS has different datasets (men, women, children, birth, and household datasets). For this study, we used the child data set (KR file). In the KR file, all children who were born in the last five years preceding the survey in the selected enumeration area were included. The DHS used two stages of stratified sampling technique to select the study participants. We pooled the DHS surveys conducted in the 12 East African countries and a total weighted sample of 79744 under-five children was included in the study.

### Variables of the study

The outcome variable was stunted (chronic undernutrition) or not stunted in children aged 6–59 months. In DHS to assess whether a child was stunted or not, the height for age measurement status was used. Children with a height for age measurement of < -2 standard deviation from the median of the reference population were considered to be short for their age (stunted), and children with a measurement of $\geq 2$ standard deviation units were considered as not stunted. The response variable for the i[th] child is represented by a random variable Yi with two possible values coded as 1 and 0. So, the response variable of the i[th] child Yi was measured as a dichotomous variable with possible values Yi = 1, if the child was stunted and Yi = 0 if a child was not stunted.

The independent variables were classified into four themes as community-level, maternal, household, and child-related variables. Community-level factors included were the place of residence, country, and distance to the health facility. The maternal factors included were maternal age, maternal education, marital status, maternal Body Mass Index (BMI), woman's health care decision making autonomy, maternal occupation, place of delivery, the mode of delivery, Antenatal Care (ANC) visit during pregnancy, Postnatal Care (PNC), media exposure, paternal education, and maternal height. Child-related variables included the sex of the

child, age of the child, type of birth, birth order, child-size at birth, exclusive breastfeeding for six months, and nutritional status (wasting, and underweight). The household factors included covered by health insurance, type of toilet, source of fuel for cooking, wealth status, sex of household head, and family size.

Media exposure was calculated by aggregating three variables such as watching television, listening to the radio, and reading newspapers. Then categorised as having media exposure if a mother has been exposed to at least one of the three and not if she had no exposure to any of the media sources.

## Data management and analysis

We pooled the DHS data from the 12 East African countries together after extracting the variables based on the literature. Before any statistical analysis, the data were weighted using sampling weight, primary sampling unit, and strata to restore the representativeness of the survey and to account for sampling design when calculating standard errors and reliable estimates. "Svy set" STATA command was used for analysis to take into account the complex survey design. Cross tabulations and summary statistics were done using STATA version 14 software. The pooled prevalence of stunting across countries was presented in a forest plot. The DHS data had a hierarchical nature, this could violate the independence of observations and equal variance assumption of the traditional logistic regression model. Hence, children are nested within a cluster and we expect that children within the same cluster are more likely to be related to each other than children in another cluster. This implies that there is a need to take into account the between cluster variability by using advanced models. Therefore, for the associated factors, we used the mixed-effect logistic regression analysis method. Model comparison and fitness were assessed based on the Intra-class Correlation Coefficient (ICC), Likelihood Ratio (LR) test, Median Odds Ratio (MOR), and deviance (-2LLR) values since the models were nested. Accordingly, a mixed effect logistic regression model (both fixed and random effect) was the best-fitted model since it had the lowest deviance value. Variables with a p-value <0.2 in the bi-variable analysis were considered in the multivariable mixed-effect logistic regression model. Adjusted Odds Ratios (AOR) with a 95% Confidence Interval (CI) and p-value ≤ 0.05 in the multivariable model were used to declare significant factors associated with stunting.

## Ethics considerations

Ethical approval and participant consent were not necessary for this particular study since the study was a secondary data analysis based on the publically available DHS data. We requested the data from the MEASURE DHS Program and permission was granted to download and use the data for this study. There are no names of individuals or household addresses in the data files.

## Results

### Characteristics of the study population

A total of 79744 under-five children were included in the study, of these 40171 (50.4%) were males. The median (±IQR) age of children was 31 (±13.5) months. About 19.7% of the children were from Kenya and the majority (76.2%) of the children were living in rural areas. The majority of the mothers had attained a primary level of education and 49.4% were aged 25–34 years. The majority (68.6%) of children were delivered at a health facility (Table 1).

**Table 1. Maternal, child, household and community level characteristics of the study population in East Africa (N = 79744).**

| Variables | Weighted frequency (%) |
|---|---|
| **Community level variables** | |
| **Country** | |
| Burundi | 5580 (7.0) |
| Comoros | 2221 (2.8) |
| Ethiopia | 8455 (10.6) |
| Kenya | 15705 (19.7) |
| Madagascar | 4522 (5.7) |
| Malawi | 4635 (5.8) |
| Mozambique | 8818 (11.1) |
| Rwanda | 3258 (4.1) |
| Tanzania | 7782 (9.8) |
| Uganda | 3881 (4.9) |
| Zambia | 10259 (12.8) |
| Zimbabwe | 4624 (5.8) |
| **Residence** | |
| Rural | 60751 (76.2) |
| Urban | 18993 (23.8) |
| **Distance to health facility** | |
| Not a big problem | 47147 (59.1) |
| A big problem | 32597 (40.9) |
| **Maternal related characteristics** | |
| **Mothers age** | |
| 15–24 | 21556 (27.0) |
| 25–34 | 39398 (49.4) |
| 35–49 | 18790 (23.6) |
| **Maternal education status** | |
| No | 20178 (25.3) |
| Primary | 41170 (51.6) |
| Secondary and higher | 18396 (23.1) |
| **Marital status** | |
| Never married | 59331 (74.4) |
| Married/living together | 11337 (14.2) |
| Divorced/widowed/separated | 9076 (11.4) |
| **Working status** | |
| No | 27941 (35.0) |
| Yes | 51803 (65.0) |
| **Maternal BMI** | |
| Underweight | 15834 (19.9) |
| Normal | 49750 (62.4) |
| Overweight | 14160 (17.7) |
| **Women health care decision making autonomy** | |
| Respondent alone | 14389 (18.0) |
| Jointly with husband/partner | 31519 (39.5) |
| Husband/partner only | 33836 (42.5) |
| **Media exposure** | |
| No | 26221 (32.9) |

(*Continued*)

**Table 1.** (Continued)

| Variables | Weighted frequency (%) |
|---|---|
| Yes | 53523 (67.1) |
| **Husband education status** | |
| No | 14503 (18.2) |
| Primary | 32181 (40.4) |
| Secondary and above | 33060 (41.4) |
| **Maternal height** | |
| Short | 9371 (11.8) |
| Normal | 70373 (88.2) |
| **Place of delivery** | |
| Home | 25055 (31.4) |
| Health facility | 54689 (68.6) |
| **Mode of delivery** | |
| Vaginal | 75240 (94.3) |
| Caesarean | 4504 (5.7) |
| **ANC visit during pregnancy** | |
| No | 28566 (35.8) |
| Yes | 51177 (64.2) |
| **PNC visit (n = 41430)** | |
| No | 23295 (56.2) |
| Yes | 18135 (43.8) |
| **Household level characteristics** | |
| **Type of toilet facility** | |
| Improved | 29686 (37.2) |
| Not improved | 50058 (62.8) |
| **Source of fuel** | |
| Modern fuel | 5137 (6.4) |
| Traditional fuel | 74607 (93.6) |
| **Covered by health insurance** | |
| No | 63723 (79.9) |
| Yes | 16021 (20.1) |
| **Wealth status** | |
| Poor | 36630 (45.9) |
| Middle | 15670 (19.7) |
| Rich | 27444 (34.4) |
| **Sex of household head** | |
| Male | 61510 (77.1) |
| Female | 18234 (22.9) |
| **Family size** | |
| 1–4 | 23288 (29.2) |
| 5–8 | 44878 (56.3) |
| $\geq 9$ | 11578 (14.5) |
| **Child characteristics** | |
| **Sex of child** | |
| Male | 40171 (50.4) |
| Female | 39573 (49.6) |
| **Age of child (in months)** | |
| < 24 | 28195 (35.3) |

(*Continued*)

**Table 1.** (Continued)

| Variables | Weighted frequency (%) |
|---|---|
| 24–35.9 | 17524 (22.0) |
| 36–47.9 | 17295 (21.7) |
| 48–60 | 16730 (21.0) |
| **Type of birth** | |
| Single | 77558 (97.3) |
| Twin | 2186 (2.7) |
| **Birth order** | |
| First | 17399 (21.8) |
| 2–4 | 38958 (48.9) |
| ≥ 4 | 23387 (29.3) |
| **Exclusively breast feed for 6 months** | |
| No | 69273 (86.7) |
| Yes | 10471 (13.2) |
| **Child size at birth** | |
| Small | 20114 (25.2) |
| Average | 36513 (45.8) |
| Large | 23117 (29.0) |
| **Wasting status** | |
| Normal | 76044 (95.4) |
| Wasted | 3700 (4.6) |
| **Underweight status** | |
| Normal | 15767 (19.8) |
| Underweight | 63977 (80.2) |

## Prevalence of stunting among under-five children in East Africa

The pooled prevalence of stunting among under-five children in East Africa was 33.3% (95% CI: 32.9, 35.6). The prevalence was varied across countries, it was highest in Burundi (53%) and lowest in Kenya (21.9%) (Fig 1).

## Associated factors of stunting among under-five children

The mixed-effect logistic regression model was the best-fitted model for the data because of the smallest value of deviance (Table 2). Furthermore, the ICC value in the null model was 18.9% [95% CI: 15.3%, 23.3%], it showed that about 18.9% of the total variability of stunting among under-five children was attributed to the between cluster variability whereas the remaining 81.1% of the total variability was explained by the between-individual variation. The MOR was 1.61 and showed that if we randomly select two children from two different clusters, a child from a cluster with a high risk of stunting was 1.61 times more likely to be stunted than a child from the cluster with a lower risk of stunting. Furthermore, the likelihood ratio test was (LR test vs. Logistic model: chibar2 (01) = 6623.18, Prob > = chiba2 = <0.001) significant which informed that the mixed-effect logistic regression model (GLMM) is the better model over the basic model (Table 2).

In the multivariable mixed-effect logistic regression analysis; residence, country, maternal education, wealth status, child age, sex of the child, child-size at birth, type of birth, place of delivery, and birth order were significantly associated with increased odds of stunting among under-five children. Whereas, maternal age, ANC visit, and family size were significantly

**Fig 1. The prevalence of stunting among under-five children in East African countries.**

associated with decreased odds of stunting among under-five children. The odds of stunting among children living in the rural area were 1.11 (AOR = 1.11, 95% CI: 1.06, 1.16) times higher compared to those children in urban areas. Under-five children in Kenya had higher odds of stunting than children in other East African countrieso. Children born to mothers who had no formal education and primary education had 1.42 (AOR = 1.42, 95% CI: 1.34, 1.50) and 1.37 (AOR = 1.37, 95% CI: 1.31, 1.44) times higher likelihood to be stunted than children born to mothers who attained secondary education and above, respectively.

Children born to poor wealth households and middle wealth households had 1.66 (AOR = 1.66, 95% CI: 1.58, 1.74), and 1.42 (AOR = 1.42, 95% CI: 1.35, 1.49) times higher odds of stunting compared to children born to the rich wealth household, respectively. Children who were 2nd -4th born and above 4th born had 1.08 (AOR = 1.08, 95% CI: 1.03, 1.13), and 1.27 (AOR = 1.27, 95% CI: 1.19, 1.35) times more likely to be stunted as compared to the first birth. Children born at home were 1.09 (AOR = 1.09, 95% CI: 1.05, 1.13) times more likely to be stunted compared to children born at a health facility.

Children aged 36–47.9 months were 1.09 times (AOR = 1.09, 95% CI: 1.04, 1.14) higher odds of stunting compared to children aged < 24 months. Being male increased the odds of stunting by 19% (AOR = 1.19, 95% CI: 1.15, 1.23) compared to female children. The odds of stunting among multiple births were 1.98 times (AOR = 1.98, 95% CI: 1.81, 2.17) higher than singletons. Children who were small size at birth were 1.35 (AOR = 1.35, 95% CI: 1.29, 1.40) times more likely to be stunted compared to those children who were average size at birth

**Table 2. Model comparison and model fitness.**

| Parameter | Standard logistic regression model | Null model | Mixed-effect logistic regression model |
|---|---|---|---|
| LLR | -47182.02 | | -47117.89 |
| Deviance | 94364.04 | | 94235.6 |
| ICC | | 18.9% (95% CI: 15.3%, 23.3%) | |
| MOR | | 1.61 (1.57, 1.65) | |
| LR-test | LR test vs. logistic model: chibar2(01) = 245.15 Prob > = chibar2 = 0.001 | | |

ICC: Intra-class Correlation Coefficient, LLR: Log-likelihood Ratio, LR: Likelihood Ratio, MOR: Median Odds Ratio

whereas the odds of stunting among children who were large size at birth were decreased by 15% (AOR = 0.85, 95% CI: 0.81, 0.88) compared to average size children at birth. The odds of stunting for children born to mothers aged 25–34 and ≥ 35 years were decreased by 17% (AOR = 0.83, 95% CI: 0.79, 0.86) and 24% (AOR = 0.76, 95% CI: 0.72, 0.81) compared to children born to mothers aged 15–24 years, respectively. The odds of stunting among children belonging to households with eight and above family members were decreased by 8% (AOR = 0.92, 95% CI: 0.87, 0.98). The odds of stunting among children whose mothers had ANC visits during pregnancy were decreased by 11% (AOR = 0.89, 95% CI: 0.86, 0.93) than children whose mothers didn't have ANC during pregnancy (Table 3).

## Discussion

This study provides evidence that stunting among under-five children continues to be a major public health problem in East African countries. Our study investigated the pooled prevalence and associated factors of stunting among under-five children in East African countries to understand the overall prevalence and associated factors of stunting in East Africa as well as the inter-country distribution of stunting among under-five children across the East African countries.

In this study, 33.3% of under-five children in East African countries were stunted. This is lower than the study reported in Sub-Saharan Africa [39] but higher than a study finding in Iran [42]. This could be due to East African countries being more vulnerable to food shortages because they rely on agriculture which is highly sensitive to weather and climate conditions such as temperature, precipitation, and light and extreme events, and low capacity for adaptation [43, 44]. Furthermore, stunting among under-five children is strongly associated with poverty [45, 46]. Iran is relatively wealthier and has good access to basic maternal and child health care services compared to East African countries, hence this could be the possible explanation for the decreased prevalence of stunting in Iran.

In the multivariable mixed-effect logistic regression; residence, country, maternal age, maternal education, wealth status, child age, sex of the child, type of birth, place of delivery, ANC visit, child-size at birth, birth order, and family size were significantly associated with stunting. In this study, under-five children in a rural area had higher odds of stunting than urban children. It was consistent with study findings [47, 48], this could be due to the reason that urban residents have good access to education, improved access to water and sanitation facilities that contributed to having good child nutrition. Besides, in urban areas maternal health care services such as ANC, health facility delivery, and PNC visit are available, that could raise community awareness to provide quality complementary feeding and uses child immunization services, this could contribute to the lower risk of child stunting in urban areas [49].

Children born to mothers with a lower level of education were more likely to be stunted compared to children born to mothers who attained a secondary and higher level of education. It is consistent with the study findings in Nairobi [50] and Bolivia [51]. It could be because educated mothers have good knowledge about child health and basic health care services, and an enhanced capacity to recognize childhood illness and seek treatment for their children [52]. Besides, educated women are more likely to exclusively breastfeed for 6 months and provide recommended complementary feeding compared to uneducated mothers [53].

In our study wealth status is one of the most important predictors of stunting. Children in the poor and middle household's wealth were more likely to be stunted than children in the richest household wealth. This was consistent with studies in Indonesia [54], Bangladesh [55], and Angola [56], and stunting is considered to be an indicator of food insecurity and poverty

**Table 3. The bi-variable and multivariable mixed-effect logistic regression analysis of stunting among under 5 children in East Africa.**

| Variable | Stunted | Not stunted | COR with 95% CI | AOR with 95% CI |
|---|---|---|---|---|
| **Residence** | | | | |
| Urban | 14535 | 4458 | 1 | 1 |
| Rural | 38691 | 22060 | 1.79 (1.72, 1.86) | 1.11 (1.06, 1.16)* |
| **Country** | | | | |
| Kenya | 12262 | 3443 | 1 | 1 |
| Comoros | 1597 | 3076 | 1.15 (1.03, 1.27) | 1.36 (1.22, 1.52)* |
| Ethiopia | 5379 | 3076 | 1.78 (1.68, 1.89) | 1.88 (1.75, 2.20)* |
| Burundi | 2620 | 2960 | 3.60 (3.37, 3.85) | 4.76 (4.40, 5.14)* |
| Madagascar | 2302 | 2220 | 2.95 (2.74, 3.17) | 3.44 (3.18, 3.72)* |
| Malawi | 3175 | 1460 | 1.48 (1.37, 1.59) | 1.89 (1.74, 2.05)* |
| Mozambique | 5361 | 3457 | 1.88 (1.77, 2.00) | 2.51 (2.34, 2.70)* |
| Rwanda | 2113 | 1145 | 1.77 (1.63, 1.93) | 2.41 (2.20, 2.64)* |
| Tanzania | 5410 | 2372 | 1.42 (1.33, 1.51) | 1.78 (1.66, 1.91)* |
| Uganda | 2943 | 939 | 1.09 (1.01, 1.19) | 1.30 (1.19, 1.42)* |
| Zambia | 6475 | 3784 | 1.93 (1.82, 2.04) | 2.43 (2.28, 2.59)* |
| Zimbabwe | 3588 | 1036 | 0.89 (0.82, 0.97) | 1.45 (1.32, 1.59)* |
| **Maternal age** | | | | |
| 15–24 | 14032 | 7523 | 1 | 1 |
| 25–34 | 26785 | 12613 | 0.91 (0.88, 0.95) | 0.83 (0.79, 0.86)* |
| ≥ 35 | 12409 | 6381 | 0.98 (0.94, 1.02) | 0.76 (0.72, 0.81)* |
| **Maternal education** | | | | |
| No | 11791 | 8387 | 2.40 (2.29, 2.51) | 1.42 (1.34, 1.50)** |
| Primary | 27003 | 14167 | 1.90 (1.83, 1.98) | 1.37 (1.31, 1.44)* |
| Secondary and above | 14431 | 3964 | 1 | 1 |
| **Wealth status** | | | | |
| Poor | 22149 | 14481 | 2.04 (1.97, 2.11) | 1.66 (1.58, 1.74)** |
| Middle | 10256 | 5413 | 1.67 (1.60, 1.75) | 1.42 (1.35, 1.49)* |
| Rich | 20820 | 6624 | 1 | 1 |
| **Media exposure** | | | | |
| Yes | 37494 | 16028 | 1 | 1 |
| No | 17732 | 10489 | 1.52 (1.47, 1.56) | 1.04 (0.99, 1.08) |
| **Women health care decision making autonomy** | | | | |
| Respondent alone | 9640 | 4749 | 1 | 1 |
| Jointly with husband/ partner | 20759 | 10760 | 1.06 (1.02, 1.11) | 0.99 (0.94, 1.03) |
| Husband/partner only | 22823 | 11008 | 1.03 (0.99, 1.08) | 1.04 (0.99, 1.09) |
| **Distance to health facility** | | | | |
| Not a big problem | 32729 | 14417 | 1 | 1 |
| A big problem | 20496 | 12101 | 1.33 (1.29, 1.37) | 1.02 (0.98, 1.06) |
| **Childs age in months** | | | | |
| <24 | 19061 | 9134 | 1 | 1 |
| 24–35.9 | 11810 | 5714 | 1.03 (0.99, 1.08) | 1.01 (0.97, 1.06) |
| 36–47.9 | 11284 | 6011 | 1.12 (1.07, 1.16) | 1.09 (1.04, 1.14)* |
| 48–60 | 11070 | 5660 | 1.08 (1.03, 1.12) | 1.02 (0.97, 1.07) |
| **Sex of child** | | | | |
| Male | 26127 | 14043 | 1.15 (1.12, 1.19) | 1.19 (1.15, 1.23)* |
| Female | 27098 | 12475 | 1 | 1 |
| **Type of birth** | | | | |

(*Continued*)

**Table 3.** (Continued)

| Variable | Stunted | Not stunted | COR with 95% CI | AOR with 95% CI |
|---|---|---|---|---|
| Single | 52082 | 25475 | 1 | 1 |
| Multiple | 1143 | 1044 | 2.04 (1.87, 2.22) | 1.98 (1.81, 2.17)** |
| **Place of delivery** | | | | |
| Health facility | 37519 | 17168 | 1 | 1 |
| Home | 15707 | 9348 | 1.29 (1.25, 1.34) | 1.09 (1.05, 1.13)* |
| **ANC visit during pregnancy** | | | | |
| No | 17874 | 10692 | 1 | 1 |
| Yes | 35352 | 15826 | 0.75 (0.73, 0.78) | 0.89 (0.86, 0.93)* |
| **Exclusively breast feed for 6 months** | | | | |
| No | 6941 | 3530 | 1 | 1 |
| Yes | 46285 | 22988 | 0.92 (0.88, 0.96) | 0.95 (0.89, 1.01) |
| **Child size at birth** | | | | |
| Average | 24347 | 12166 | 1 | 1 |
| Small | 13104 | 7010 | 1.12 (1.08, 1.16) | 1.35 (1.29, 1.40)** |
| Large | 15775 | 7342 | 0.89 (0.86, 0.92) | 0.85 (0.81, 0.88)** |
| **Birth order** | | | | |
| First | 11987 | 5412 | 1 | 1 |
| 2–4 | 26557 | 12401 | 1.06 (1.02, 1.10) | 1.08 (1.03, 1.13)** |
| ≥ 5 | 14682 | 8706 | 1.33 (1.28, 1.39) | 1.27 (1.19, 1.35)* |
| **Family size** | | | | |
| 1–4 | 15871 | 7416 | 1 | 1 |
| 5–8 | 29520 | 15358 | 1.15 (1.11, 1.19) | 1.03 (0.99, 1.07) |
| >8 | 7835 | 3744 | 1.05 (1.01, 1.10) | 0.92 (0.87, 0.98)* |

* p-value<0.05,

** p-value<0.01, CI: Confidence interval, COR: Crude Odds Ratio, AOR: Adjusted Odds Ratio

This table includes only those variables for which the bi-variable analysis had a p <0.20

[57]. Children who live in poor households typically have poor access to adequate food, safe water, and sanitation [58]. Consequently, they are at higher risk of childhood infectious diseases such as acute respiratory diseases, diarrheal diseases, and intestinal parasites, all of which contribute to chronic undernutrition [59].

Children in the age group of 36–48 months were more likely to be stunted compared to children aged less than 24 months. It is consistent with a study reported in Nigeria [60], this could be due to the reason that as children ages increase they had environmental exposure that could increase their risk of infections like tonsillopharyngitis, diarrheal diseases, pneumonia, and CROUP [61, 62]. The risk of malnutrition is highest mainly after the initiation of complementary feeding that has been directly linked with poor complementary feeding and breastfeeding weaning practices of mothers, together with high rates of infectious diseases could be the possible reason for the increased risk of stunting in older children [63, 64].

Males were more likely to be stunted than females. This is supported by previous study findings reported in sub-Saharan Africa [65], Pakistan [66], and Brazil [67]. This could be due to the slower lung maturation among males compared to females that predispose male children to repeated respiratory infections such as pneumonia, bronchiolitis, otitis media, and hyperactive airway disease which could contribute to the increased risk of stunting among males [68]. Children born with multiple births are at higher risk of being stunted compared to singletons. It is consistent with studies reported in India [69] and Nigeria [40] and the

increased risk of stunting is likely explained by the fact that multiple births are more likely to be born prematurely with low birth weight, and experience increased competition for nutrition [70].

In our study, birth order and, family size were significant predictors of stunting among under-five children. Children of two and above birth order were more likely to be stunted than children of first birth order, it was in line with previous study findings [71, 72]. This might be due to the reason that the family unable to satisfy child dietary and other healthcare-related services because of more children as well as due to maternal nutrition depletion [73].

Children who were small size at birth were more likely to be stunted whereas large-sized babies at birth had lower odds of being stunted as compared to children who were average size at birth. It was consistent with studies [74, 75], this could be due to increased vulnerability of children with small size babies to infections mainly diarrheal and lower respiratory infections such as pneumonia, and otitis media, and increased risk of complications including sleep apnea, anemia, chronic lung disorders, fatigue and loss of appetite compared to children with normal birth weights [76, 77].

The odds of stunting for children born to mothers aged 25–34 and $\geq 35$ years were lower than children born to mothers aged 15–24 years. It was in line with the study reported in Nigeria [60]. The possible explanation could be due to the reason that teenagers who gave birth are less likely to use health services like vaccination services, and have poor health-seeking behavior relative to mature mothers.

In our study, health facility delivery and ANC visit were protective of child chronic undernutrition. This finding is supported by studies reported in Indonesia [78] and Nepal [79], this could be due to the reason that ANC visit is an entry point for the other maternal health services, and births from mother who had ANC visit and health facility delivery are aware of danger signs of childhood illness, using childhood immunization services and appropriate childhood feeding practice [80].

The strength of this study was that it was based on a weighted large, nationally representative data set and could have adequate statistical power to detect the true association of factors with stunting among under-five children. Besides, the study is done using an advanced model to take into account the clustering effect (mixed-effect logistic regression) to get reliable standard error and estimate. However, the study finding is interpreted in light of limitations. First, as with other cross-sectional studies, the temporal relationship can't be established. Second, the DHS didn't incorporate information about health care availability and accessibility like distance to the health facility, medical-related factors, and the quality of maternal health services provided which might influence child nutrition status. Also, since data was collected from self-report from respondents there may be a possibility of social desirability and recall bias.

## Conclusion

The pooled prevalence of stunting among under-five children in East Africa was 33.3% ranging from 21.9% in Kenya to 53% in Burundi. Chronic child undernutrition continues to be a major public health problem in East Africa. In this study; residence, country, maternal age, maternal education, wealth status, child age, sex of the child, type of birth, place of delivery, ANC visit, child-size at birth, birth order, and family size were significantly associated with stunting among under 5 children. Therefore, to improve child nutrition status the governmental and non-governmental organizations should design public health interventions targeting rural residents, and the poorest households. Furthermore, enhancing health facility delivery, ANC visit, and maternal health education is vital for reducing child chronic undernutrition.

## Acknowledgments

We greatly acknowledge MEASURE DHS for granting access to the East African DHS data sets.

## Author Contributions

**Conceptualization:** Getayeneh Antehunegn Tesema, Yigizie Yeshaw, Misganaw Gebrie Worku, Zemenu Tadesse Tessema, Achamyeleh Birhanu Teshale.

**Data curation:** Getayeneh Antehunegn Tesema, Yigizie Yeshaw, Misganaw Gebrie Worku, Zemenu Tadesse Tessema, Achamyeleh Birhanu Teshale.

**Formal analysis:** Getayeneh Antehunegn Tesema, Misganaw Gebrie Worku, Zemenu Tadesse Tessema, Achamyeleh Birhanu Teshale.

**Investigation:** Getayeneh Antehunegn Tesema, Yigizie Yeshaw, Misganaw Gebrie Worku, Zemenu Tadesse Tessema, Achamyeleh Birhanu Teshale.

**Methodology:** Getayeneh Antehunegn Tesema, Yigizie Yeshaw, Misganaw Gebrie Worku, Zemenu Tadesse Tessema, Achamyeleh Birhanu Teshale.

**Software:** Getayeneh Antehunegn Tesema, Yigizie Yeshaw, Misganaw Gebrie Worku, Zemenu Tadesse Tessema, Achamyeleh Birhanu Teshale.

**Supervision:** Getayeneh Antehunegn Tesema, Yigizie Yeshaw, Misganaw Gebrie Worku, Zemenu Tadesse Tessema, Achamyeleh Birhanu Teshale.

**Validation:** Getayeneh Antehunegn Tesema, Yigizie Yeshaw, Misganaw Gebrie Worku, Zemenu Tadesse Tessema, Achamyeleh Birhanu Teshale.

**Visualization:** Getayeneh Antehunegn Tesema, Yigizie Yeshaw, Misganaw Gebrie Worku, Zemenu Tadesse Tessema, Achamyeleh Birhanu Teshale.

**Writing – original draft:** Getayeneh Antehunegn Tesema, Yigizie Yeshaw, Misganaw Gebrie Worku, Zemenu Tadesse Tessema, Achamyeleh Birhanu Teshale.

**Writing – review & editing:** Getayeneh Antehunegn Tesema, Yigizie Yeshaw, Misganaw Gebrie Worku, Achamyeleh Birhanu Teshale.

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
