## [Decision Letter · Decision Letter 0]

18 Jan 2021

PONE-D-20-18977

Pooled prevalence and associated  factors of chronic undernutrition among under-five children in East Africa: A multilevel analysis

PLOS ONE

Dear Dr. Tesema,

Thank you for submitting your manuscript to PLOS ONE. After careful consideration, we feel that it has merit but does not fully meet PLOS ONE’s publication criteria as it currently stands. Therefore, we invite you to submit a revised version of the manuscript that addresses the points raised during the review process.

I have reviewed the paper as the second reviewer and have identified a number of issues to be addressed in your revision.  In addition, I have copy edited the paper to improve the English language and writing style. These changes are made to a copy of the paper which is attached to this letter.

We look forward to receiving your revised manuscript.

Kind regards,

Jane Anne Scott, PhD, MPH Grad Dip Dietetics, BSc

Academic Editor

PLOS ONE

Journal Requirements:

4. We noticed you have some minor occurrence of overlapping text with the following previous publication(s), which needs to be addressed:

https://link.springer.com/article/10.1007/s10668-010-9278-0?code=17828610-9ce7-412d-b01f-238f1863e02f&error=cookies_not_supported

https://bmcpublichealth.biomedcentral.com/articles/10.1186/s12889-020-09965-y

https://journals.plos.org/plosone/article?id=10.1371/journal.pone.0157814

In your revision ensure you cite all your sources (including your own works), and quote or rephrase any duplicated text outside the methods section. Further consideration is dependent on these concerns being addressed.

Reviewers' comments:

Reviewer's Responses to Questions

**Comments to the Author**

1. Is the manuscript technically sound, and do the data support the conclusions?

Reviewer #1: Yes

Reviewer #2: Yes

2. Has the statistical analysis been performed appropriately and rigorously? 

Reviewer #1: Yes

Reviewer #2: Yes

3. Have the authors made all data underlying the findings in their manuscript fully available?

Reviewer #1: Yes

Reviewer #2: Yes

4. Is the manuscript presented in an intelligible fashion and written in standard English?

Reviewer #1: Yes

Reviewer #2: No

5. Review Comments to the Author

Reviewer #1: The study is relevant and timeous considering that stunting remains persistent in most African countries. The authors determined the pooled prevalence of chronic undernutrition and associated factors. Data from 12 East African DHS was used (i.e. secondary analysis). The overall pooled prevalence for stunting and by country are reported. Regression analysis was used to study the associated factors, and the manuscript report several factors similar to other studies, such as residence, maternal age, maternal education, wealth status, birth size, sex of the child, ANC visit, place of delivery, family size, type of birth, birth order, country, and child age. The strength and the limitations of the study are well discussed and the conclusion summarizes the findings and the recommendation.

The manuscript has adhered to the standard format.

Abstract

Under the results; the third sentence should read "Children whose mothers" not mother.

Key words

The authors can consider to add stunting, pooled prevalence, associated factors, under-five children, East-Africa, DHS . Remove "mixed effect analysis".

Background

first paragraph sentence 7 - A child is considered stunted... The authors should remove "as"

second paragraph this sentence is missing something and should be rephrased "Malnutrition is the leading cause of under-five morbidity and mortality". Is it under-five 'children'?

Results

This sentence "The median age of children was 31 (IQR±13.5) months" should either report median (IQR) or mean±SD. It cant be IQR and SD.

Reviewer #2: In general, this is a well conducted study although the writing is awkward in places with some grammatical errors. A lack of line numbering made it difficult to highlight these issues so I have edited the paper and I attach a copy of this paper with changes marked. Further suggestions for improvements are listed below.

Abstract

1. The results section of the abstract contains a single, very long sentence which contains all of the factors significantly associated with undernutrition. I suggest breaking this into two sentences which first identifies those factors associated with an increased risk of stunting and then a second sentence which identifies those factors which were associated with a decreased risk of stunting.

Background

2. Is the statistic of 1 in 4 infants in East African countries being stunted actually for infants (i.e. children under 12 months of age) or for children under-five years, which is the target group for this study?

Methods

3. The explanatory variable ‘exclusive breastfeeding’ needs to be more clearly defined. Is this a continuous variable which indicates the duration of EBF or is it a binary variable which indicates whether a child was EBF to 6 months of age? The results in table 4 suggest it is the latter.

Results

4. Under the data management section you say that the pooled prevalence of stunting with the 95% Confidence Interval (CI) was reported using a forest plot. However, while you report the pooled prevalence in the results section you have not included the forest plot.

5. When reporting the findings from the multivariable mixed-effect logistic regression analysis you describe each variable individually and you move between variables that are associated with an increased risk of stunting and those with a decreased risk of stunting. I think it would be clearer if you were to first present those variables associated with a higher risk of stunting and then in a new paragraph identify those that are associated with a lower risk of stunting.

6. The sentence that compares the odds of stunting for each country compared to Kenya is very long and repeats results that are easily discernible from the table. I suggest simplifying this finding and only provide the results for the countries with the lowest and highest odds ratios e.g. “Compared with Kenya which had the lowest prevalence of stunting, all of the other countries had a significantly higher odds of stunting ranging from an odds of 1.30 (1.19, 1.42) for Uganda to 4.76 (4.40, 5.14) for Burundi.”

7. PNC is described in the methods section as an explanatory variable. While it is reported in table 1 there is no further reference to PNC. This variable isn’t included in Table 3 or reported in the results or discussion in terms of its association with stunting. You should either remove all reference to PNC or redo the analysis with this variable in the model. Or does table 3 only contain those variables which had a p-value above 0.2 in the bivariable analysis? This is not clear if this is the case.

Discussion

8. For clarity, I suggest discussing all of the factors associated with an increased risk of stunting and THEN discussing those factors associated with a decreased risk. You move between factors associated with increased and decreased risk and then back to increased risk.

References

9. A number of the references are very dated e.g. 1, 11, 16, 32, 41, 52, 57, 61, 65. Are they seminal papers or can they be replaced with more recent references?

10. Reference 7 The last ‘author’ appears to be a group/organisation but it is unclear which group. When entering the details of a group put a comma after the last word of the name to denote that this is a group. E.g.

WHO Child Health Epidemiology Reference Group,

Otherwise ENDNOTE treats the last word as the family name and initialises all of the other words e.g. Group,WCHERG

11. Reference 33 Some details appear to be missing. Is this a book or a chapter of a book?

12. Reference 35 Please do not abbreviate the institution which has published the report. Where was this report published?

13. Reference 59 Reference details are incomplete.

6. PLOS authors have the option to publish the peer review history of their article (what does this mean?). If published, this will include your full peer review and any attached files.

Reviewer #1: No

Reviewer #2: **Yes: **Jane A Scott

---

## [Author Response · Author response to Decision Letter 0]

10 Feb 2021

PLOS ONE 

Point by point response for editors/reviewers comments 

Manuscript title: Pooled prevalence and associated factors of chronic undernutrition among under-five children in East Africa: A multilevel analysis

Manuscript ID: PONE-D-20-18977

Dear editor/reviewer. 

Dear all,

We would like to thank you for these constructive, building, and improvable comments on this manuscript that would improve the substance and content of the manuscript. We considered each comment and reviewers on the manuscript thoroughly. Our point-by-point responses for each comment and question are described in detail on the following pages.

Response to reviewers comments

Reviewer#1

1. Abstract

Under the results; the third sentence should read "Children whose mothers" not mother.

Authors’ response: Thank you for the comments. We accept and modified it. (See Abstract section, line 41, page 2)

2. Key words

The authors can consider to add stunting, pooled prevalence, associated factors, under-five children, East-Africa, DHS . Remove "mixed effect analysis".

Authors’ response: Thank you for the concerns. We removed it in the revised manuscript. (See Abstract section, line 57, line 3)

3. Background

first paragraph sentence 7 - A child is considered stunted... The authors should remove "as"

second paragraph this sentence is missing something and should be rephrased "Malnutrition is the leading cause of under-five morbidity and mortality". Is it under-five 'children'?

Authors’ response: Thank you for the comments. We accepted and modified it. (See Background section, page 4)

4. Results

This sentence "The median age of children was 31 (IQR±13.5) months" should either report median (IQR) or mean±SD. It cant be IQR and SD.

Authors’ response: Thank you for the comments. We reported the median as a measure of central tendency and Inter-quartile Rage (IQR) as a measure of dispersion since the variable was not normally distributed (skewed) with Shapiro Wilks test p-value<0.05.

Reviewer #2

1. In general, this is a well conducted study although the writing is awkward in places with some grammatical errors. A lack of line numbering made it difficult to highlight these issues so I have edited the paper and I attach a copy of this paper with changes marked. Further suggestions for improvements are listed below.

Authors’ response: Thank you reviewer for the detailed comments and changes you made for the betterment of the paper. Sorry for the missing line number, and now, we insert the line number and address the comments you raised. (See the revised manuscript)

2. Abstract

The results section of the abstract contains a single, very long sentence which contains all of the factors significantly associated with undernutrition. I suggest breaking this into two sentences which first identifies those factors associated with an increased risk of stunting and then a second sentence which identifies those factors which were associated with a decreased risk of stunting.

Authors’ response: Thank you for the comments. We accept the comments and write accordingly. (See Abstract section, line 39 -52, page 2-3)

3. Background

Is the statistic of 1 in 4 infants in East African countries being stunted actually for infants (i.e. children under 12 months of age) or for children under-five years, which is the target group for this study?

Authors’ response: Thank you for the comments. We rewrote it as our target population is under-five children. (See the revised manuscript)

4. Methods

The explanatory variable ‘exclusive breastfeeding’ needs to be more clearly defined. Is this a continuous variable which indicates the duration of EBF or is it a binary variable which indicates whether a child was EBF to 6 months of age? The results in table 4 suggest it is the latter.

Authors’ response: Thank you for the comments. We consider the duration of EBF as a binary outcome by categorizing those children who breastfeed for a minimum of 6 months exclusively as yes and for those children who feed less than months as no. (See the revised manuscript)

5. Results

5. Under the data management section you say that the pooled prevalence of stunting with the 95% Confidence Interval (CI) was reported using a forest plot. However, while you report the pooled prevalence in the results section you have not included the forest plot.

Authors’ response: Thank you for the comments. We planned to present using a forest plot but we prefer to present it in a bar graph as this study was not a metanalysis. As you know while we have done, in the forest plot several columns were presented like weight but this may not be important to present it. So, we presented the pooled prevalence in the bar graph. (See Figure 1)

6. When reporting the findings from the multivariable mixed-effect logistic regression analysis you describe each variable individually and you move between variables that are associated with an increased risk of stunting and those with a decreased risk of stunting. I think it would be clearer if you were to first present those variables associated with a higher risk of stunting and then in a new paragraph identify those that are associated with a lower risk of stunting

Authors’ response: Thank you for the comments. We accept the comments and report the findings from factors that increase the odds of stunting to factors that decrease the odds of stunting. (See the Result section, line 195 – 228, page 9-11)

7. The sentence that compares the odds of stunting for each country compared to Kenya is very long and repeats results that are easily discernible from the table. I suggest simplifying this finding and only provide the results for the countries with the lowest and highest odds ratios e.g. “Compared with Kenya which had the lowest prevalence of stunting, all of the other countries had a significantly higher odds of stunting ranging from an odds of 1.30 (1.19, 1.42) for Uganda to 4.76 (4.40, 5.14) for Burundi.”

Authors’ response: Thank you for the comments. We revised it. (See Result section, line 202-203, page 10)

8. PNC is described in the methods section as an explanatory variable. While it is reported in table 1 there is no further reference to PNC. This variable isn’t included in Table 3 or reported in the results or discussion in terms of its association with stunting. You should either remove all reference to PNC or redo the analysis with this variable in the model. Or does table 3 only contain those variables which had a p-value above 0.2 in the bivariable analysis? This is not clear if this is the case.

Authors’ response: Thank you for the concern. We consider PNC in the method and result section. But, in the multivariable multilevel analysis, we did not use PNC as it has a p-value>0.2 in the bivariable analysis. As we stated in the method section we consider variables with a p-value in the bivariable multilevel analysis were included in the multivariable multilevel analysis, that is why we did not include PNC in the final model.

9. Discussion

For clarity, I suggest discussing all of the factors associated with an increased risk of stunting and THEN discussing those factors associated with a decreased risk. You move between factors associated with increased and decreased risk and then back to increased risk.

Authors’ response: Thank you for the suggestions. We accept it and rewrote it. (See discussion section)

10. References

 A number of the references are very dated e.g. 1, 11, 16, 32, 41, 52, 57, 61, 65. Are they seminal papers or can they be replaced with more recent references?

 Reference 7 The last ‘author’ appears to be a group/organisation but it is unclear which group. When entering the details of a group put a comma after the last word of the name to denote that this is a group. E.g.

WHO Child Health Epidemiology Reference Group,

Otherwise ENDNOTE treats the last word as the family name and initialises all of the other words e.g. Group,WCHERG

Reference 33 Some details appear to be missing. Is this a book or a chapter of a book?

Reference 35 Please do not abbreviate the institution which has published the report. Where was this report published?

Reference 59 Reference details are incomplete.

Authors’ response: Thank you for the comments. We modified all the suggested references. (See the revised manuscript)

---

## [Editor Report · Decision Letter 1]

23 Feb 2021

PONE-D-20-18977R1

Pooled prevalence and associated  factors of chronic undernutrition among under-five children in East Africa: A multilevel analysis

PLOS ONE

Dear Dr. Tesema,

Thank you for resubmitting your manuscript to PLOS ONE. After careful consideration, we feel that it has merit but still does not fully meet PLOS ONE’s publication criteria as it currently stands. Therefore, we invite you to submit a revised version of the manuscript that addresses the minor issues that I have identified at the end of this letter.

Please also note that your image file "Figure 1.tif" could not be opened and processed. It appears that the image file is corrupt or invalid. Please check and make sure that this problem is corrected when you resubmit your final version of the paper. 

We look forward to receiving your revised manuscript.

Kind regards,

Jane Anne Scott, PhD, MPH Grad Dip Dietetics, BSc

Academic Editor

PLOS ONE

Journal Requirements:

Additional Editor Comments (if provided):

Minor issues to be addressed

1. Line 49 Being small size at birth should be included with the factors that were associated with increased odds of stunting, not those associated with a decreased odds as the AOR is greater than 1 (AOR=1.35, 95% CI: 1.29, 1.40).

2. Line 55 How are ‘multiple births’ a modifiable factor unless a woman is having fertility treatment? The conclusions in your abstract should be consistent with the conclusions in your main paper on page 15.

3. Line 60 you have misquoted reference 3. It is the first 1000 days ‘post conception’ NOT ‘after birth’.

4. Line 304-305 The way in which you have worded this sentence makes it sound as though ‘health facility delivery and ANC visit’ are associated with an increased risk of chronic malnutrition and NOT that they are protective of chronic malnutrition. I suggest rewording this sentence

‘In our study, health facility delivery and ANC visit were protective of child chronic malnutrition.’

5. Table 3 please include a footnote explaining the meaning of the asterisks. Also I recommend that you included a footnote to explain that the table includes only those variables for which the bi-variable analysis had a p <0.20, otherwise readers may wonder why some of the variables listed in Table 1 are not included in Table 2, (as I was).

Minor grammatical errors

6. line 34 should read ‘Variables’ (plural)

7. line 37 should read ‘were reported for significant factors’

8. Line 45 2nd -4th birth order

9. line 72 the word half is a collective noun and is treated as a singular. Therefore this should read ‘More than half …… is due ….’

10. line 84 the word ‘to’ is not needed, should read ‘… children include residence…..’

11. Line 89 should read ‘ has declined from..’

12. Line 139 I suggest replacing the word ‘divided’ with ‘categorised’.

13. Line 149 should read ‘presented in a bar graph’.

14. Line 195 Given that the next paragraph describes the multivariable mixed-effect logistic regression analysis, shouldn’t this subheading read ‘The mixed effect results’?

15. Lines 260 to 262 should read ‘Besides, educated women are more likely to exclusively breastfeed for….”

16. Line 266 suggest rewording this sentence ‘Children who live in poor households typically have poor access to adequate food, safe water, and sanitation.’

---

## [Author Response · Author response to Decision Letter 1]

26 Feb 2021

Point by point response for editors/reviewers comments 

PLOS ONE Journal 

Manuscript title: Pooled prevalence and associated factors of chronic undernutrition among under-five children in East Africa: A multilevel analysis

Manuscript ID: PONE-D-20-18977R1

Dear editor/reviewer. 

Dear all,

We would like to thank you for these constructive, building, and improvable comments on this manuscript that would improve the substance and content of the manuscript. We considered each comment and clarification questions of editors and reviewers on the manuscript thoroughly. Our point-by-point responses for each comment and question are described in detail on the following pages. 

Response to Editors

Authors’ response: Thank you for the comments. We have checked the references and it is correctly presented.

Response to reviewers 

Minor issues to be addressed

1. Line 49 Being small size at birth should be included with the factors that were associated with increased odds of stunting, not those associated with a decreased odd as the AOR is greater than 1 (AOR=1.35, 95% CI: 1.29, 1.40).

Authors’ response: Thank you for the comments. We included with the factors that were associated with increased odds of stunting. (See Abstract section line 48, page 2)

2. 2. Line 55 How are ‘multiple births’ a modifiable factor unless a woman is having fertility treatment? The conclusions in your abstract should be consistent with the conclusions in your main paper on page 15

Authors’ response: Thank you for the comments. We rewrote it. (See Abstract section, line 53-57, page 3)

3. Line 60 you have misquoted reference 3. It is the first 1000 days ‘post conception’ NOT ‘after birth’

Authors’ response: Thank you for the comments. We rewrote it. (see Background section, line 61, page 4)

4. Line 304-305 The way in which you have worded this sentence makes it sound as though ‘health facility delivery and ANC visit’ are associated with an increased risk of chronic malnutrition and NOT that they are protective of chronic malnutrition. I suggest rewording this sentence

‘In our study, health facility delivery and ANC visit were protective of child chronic malnutrition.’

Authors’ response: Thank you for the suggestions. We accept it and revised the sentence. (See Discussion section, line 302-303, page 14)

5. Table 3 please include a footnote explaining the meaning of the asterisks. Also I recommend that you included a footnote to explain that the table includes only those variables for which the bi-variable analysis had a p <0.20, otherwise readers may wonder why some of the variables listed in Table 1 are not included in Table 2, (as I was).

Authors’ response: Thank you for the comments. We included the points you raised. (See Table 3)

Grammatical errors

6. line 34 should read ‘Variables’ (plural)

7. line 37 should read ‘were reported for significant factors’

8. Line 45 2nd -4th birth order

9. line 72 the word half is a collective noun and is treated as a singular. Therefore this should read ‘More than half …… is due ….’

10. line 84 the word ‘to’ is not needed, should read ‘… children include residence…..’

11. Line 89 should read ‘ has declined from..’

12. Line 139 I suggest replacing the word ‘divided’ with ‘categorised’.

13. Line 149 should read ‘presented in a bar graph’.

14. Line 195 Given that the next paragraph describes the multivariable mixed-effect logistic regression analysis, shouldn’t this subheading read ‘The mixed effect results’?

15. Lines 260 to 262 should read ‘Besides, educated women are more likely to exclusively breastfeed for….”

16. Line 266 suggest rewording this sentence ‘Children who live in poor households typically have poor access to adequate food, safe water, and sanitation.’

Authors’ response: We thank you a lot for your extensive effort to improve our work. We accept all the abovementioned comments and revised the manuscript. (See the revised manuscript)

---

## [Editor Report · Decision Letter 2]

3 Mar 2021

Pooled prevalence and associated  factors of chronic undernutrition among under-five children in East Africa: A multilevel analysis

PONE-D-20-18977R2

Dear Dr. Tesema,

We’re pleased to inform you that your manuscript has been judged scientifically suitable for publication and will be formally accepted for publication once it meets all outstanding technical requirements.

Kind regards,

Jane Anne Scott, PhD, MPH Grad Dip Dietetics, BSc

Academic Editor

PLOS ONE

---

## [Editor Report · Acceptance letter]

17 Mar 2021

PONE-D-20-18977R2 

Pooled prevalence and associated  factors of chronic undernutrition among under-five children in East Africa: A multilevel analysis 

Dear Dr. Tesema:

I'm pleased to inform you that your manuscript has been deemed suitable for publication in PLOS ONE. Congratulations! Your manuscript is now with our production department. 

Kind regards, 

on behalf of

Dr. Jane Anne Scott 

Academic Editor

PLOS ONE